# Evolution of the *WRKY* Family in Angiosperms and Functional Diversity under Environmental Stress

**DOI:** 10.3390/ijms25063551

**Published:** 2024-03-21

**Authors:** Weihuang Wu, Jinchang Yang, Niu Yu, Rongsheng Li, Zaixiang Yuan, Jisen Shi, Jinhui Chen

**Affiliations:** 1State Key Laboratory of Tree Genetics and Breeding, Research Institute of Tropical Forestry, Chinese Academy of Forestry, Guangzhou 510520, China; wh879966@163.com (W.W.); jcyang@caf.ac.cn (J.Y.); niuyu@caf.ac.cn (N.Y.); fjlrs@tom.com (R.L.); yzx0090@163.com (Z.Y.); 2State Key Laboratory of Tree Genetics and Breeding, Nanjing Forestry University, Nanjing 210037, China

**Keywords:** WRKY transcription factors, angiosperm, evolution, environmental stress

## Abstract

The transcription factor is an essential factor for regulating the responses of plants to external stimuli. The WRKY protein is a superfamily of plant transcription factors involved in response to various stresses (e.g., cold, heat, salt, drought, ions, pathogens, and insects). During angiosperm evolution, the number and function of WRKY transcription factors constantly change. After suffering from long-term environmental battering, plants of different evolutionary statuses ultimately retained different numbers of *WRKY* family members. The WRKY family of proteins is generally divided into three large categories of angiosperms, owing to their conserved domain and three-dimensional structures. The WRKY transcription factors mediate plant adaptation to various environments via participating in various biological pathways, such as ROS (reactive oxygen species) and hormone signaling pathways, further regulating plant enzyme systems, stomatal closure, and leaf shrinkage physiological responses. This article analyzed the evolution of the *WRKY* family in angiosperms and its functions in responding to various external environments, especially the function and evolution in Magnoliaceae plants. It helps to gain a deeper understanding of the evolution and functional diversity of the *WRKY* family and provides theoretical and experimental references for studying the molecular mechanisms of environmental stress.

## 1. Introduction

The transcription factor (TF), also known as the trans-acting factor, regulates gene expression by binding cis-acting elements in the upstream region of its target gene; the expression of many abiotic stress-related genes in plants is mainly regulated in this way [1,2,3,4,5]. Transcription factors can be divided into two categories according to their expression patterns: constitutive and inducible transcription factors [5,6,7,8]. Constitutive transcription factors are genes that can be expressed without environmental impact, while inducible transcription factors are genes that need specific environmental conditions to be expressed. There are various types of transcription factors in plants [2,4,8]. At present, hundreds of transcription factors have been reported in plants, and they are involved in various abiotic and biological stresses, such as high salt, drought, low temperature, hormones, and pathogenic bacteria, as well as involvement in the regulation of plant growth and development [3,4,7,9,10]. Transcription factors include ARF (auxin response factor), bHLH (basic helix-loop-helix), bZIP (basic leucine zipper), CSD (cold shock domain), HSF (heat shock transcription factor), LFY (floral meristem identity genes, LEAFY), MADS (MCM1/AG/DEF/SRF), MYB (v-myb avian myeloblastosis viral oncogene homolog), NAC (NAM/ATAF/CUC), SBP (SQUAMOSA promoter binding protein), TAZ (transcriptional coactivator with PDZ binding), TiFY (TIF[F/Y]XG), and WRKY (WRKYGQK) [2,3,9].

Plant transcription factors mainly activate or inhibit the expression of downstream genes by binding the upstream regulatory sequence of target genes and interacting with other proteins or TFs to form polymers [1,2,3,11]. Transcription factors are mainly composed of DNA-binding regions and transcriptional regulatory regions. The former determines its specific binding with the cis-acting elements of target genes, and the latter determines its regulatory role on downstream genes [5,10,11]. Transcription factors play an important role in plant stress resistance, which can regulate the expression of signal-related genes in plants that are sensitive to high temperature, low temperature, drought, high salt, and so on [1,3,7,10]. Plant transcription factors are also key regulators of plant growth, development, and morphogenesis [3,5,12]. For example, the WRKY transcription factors are mainly involved in plant stress responses, including abiotic stresses, such as high temperature, drought, low temperature, high salt, ions, and biological stresses, such as pathogens and insects [1,3,7]. 

The evolution of plants gradually transitioned from aquatic to terrestrial, transitioning from algae, mosses, ferns, and gymnosperms to angiosperms. Angiosperms are further divided into basal angiosperms, magnolias, monocotyledonous plants, and dicotyledonous plants [1,2,3,5,7]. Magnolia plants are relatively unique, with commonly *Cinnamomum kanehirae*, *Liriodendron chinense*, and *Liriodendron tulipifera* currently available. Existing reports on the evolution of angiosperms have hardly systematically studied the evolution of magnolia plants. This study analyzes the grouping, structure, origin, evolution, functional differentiation, environmental selection pressure, whole genome duplication events, growth, development, biotic stress, and abiotic stress of the *WRKY* family in angiosperms from the perspective of the overall evolution of angiosperms. This study can better and more comprehensively explain the evolution of the *WRKY* gene family in angiosperms and the functional diversity in response to external stress, providing more theoretical and experimental references for deeper research into the biological functions of *WRKY* genes and plant environmental adaptability.

## 2. Results

### 2.1. Grouping Characteristics of the WRKY Family

To analyze the evolution of the *WRKY* family from the perspective of the overall evolution of angiosperms, the *WRKY* family protein sequences of two basal angiosperms, three magnolia plants, six monocotyledonous plants, and thirteen dicotyledonous plants were selected from the “Phytozome” website to construct a phylogenetic tree of twenty-four species of angiosperms. A total of 2274 genes were classified into three major groups and seven subgroups. The second largest group, Group II, was divided into five subgroups (Figure 1). A total of 24 angiosperm species were identified, including two basal angiosperms: *Amborella trichopoda* and *Nymphaea colorata*. Three magnolias were identified: *Cinnamomum kanehirae*, *Liriodendron chinense* and *Liriodendron tulipifera*. Six monocotyledons were also identified: *Brachypodium distachyon*, *Brachypodium arbuscula*, *Oryza sativa*, *Sorghum bicolor*, *Zea mays*, *Zostera marina*. Thirteen dicotyledonous plants were studied, including *Arabidopsis thaliana*, *Carya illinoinensis*, *Coffea arabica*, *Corymbia citriodora*, *Gossypium hirsutum*, *Malus pumila*, *Cirus trifoliata*, *Populus trichocarpa*, *Portulaca amilis*, *Sinapis alba*, *Theobroma cacao*, *Vigna unguiculata* and *Vitis vinifera*.

Although the number of members of the *WRKY* family varied among the species of different evolutionary statuses, they were all divided into three major groups and seven subgroups (Table 1). The minimum number of members of the group I subgroup was seven, namely *Amborella trichopoda*, *Lirioderon chinense*, *Zostela marina*, *Oryza sativa*, and the largest, *Sinapis alba*. The subgroups II-A, II-B, II-C, II-D, and II-E of *Amborella trichopoda* had the fewest members, with two, five, six, six, and three, respectively. In contrast, *Gossypium hirsutum* had the highest number of members in the same five subgroups. The number of group II-A in *Zostela marina* was also at least two, while the number of group II-B in *Portulaca amilis* and *Brachypodium distachyon* was also at least five. The lowest number of subgroups in group III was only five in *Nymphaea colorata*, and the highest number was fifty-one in *Sinapis alba*. This result indicates that the number of subgroups in different species varied wildly, suggesting that they may have undergone environmental screening during the evolutionary process, ultimately retaining the current number of families.

### 2.2. The Structural Characteristics and Grouping of the WRKY Family

The WRKY transcription factors are mainly a family composed of one or two WRKY domains. In recent years, with the release of various plant genome data, the *WRKY* gene family has been identified within the genome, and the expression patterns of the *WRKY* genes in different plants in response to abiotic stress have been explored using qRT-PCR and RNA-seq. The verification of their functions combined with traditional molecular experimental methods has become a hotspot in botany research.

The main structure of the WRKY transcription factor is composed of four bata folds and zinc finger structures, with approximately 60 amino acids. Its motif is CX4-5CX22-23HXH (C2H2) or CX7CX23HXH (C2H2). It is highly conserved in the WRKYGQK heptapeptide region, so it is called the WRKY domain. In most WRKY domains, there are two intron insertion sites, PR and VQR. Different variants in the WRKY domain, WRRY, WSKY, WKRY, WVKY, or WKKY, replace the WRKY amino acid sequence. The zinc finger structure at the end of the folded sheet is mainly formed by the highly conservative Cys/His residues, and the N-terminal is connected via Gly to form hydrogen bonds, and then forms the β chain β folding sheet.

The WRKY family is mainly divided into three types according to the number of N-terminal conservative domains and the type of C-terminal zinc finger. The first type is the zinc finger structure containing two WRKY domains and one C2H2. The second type contains only one WRKY domain and one C2H2 zinc finger structure. The second group is further divided into five subgroups according to different amino acid sequences, namely II (a), II (b), II (c), II (d), and II (e); The third category contains only one WRKY domain, which is the same as the second category. The difference is that its zinc finger structure is C2HC, which is different from the previous two types of zinc finger structures.

Through the previous evolutionary tree analysis and species tree analysis, it was found that the evolution of the WRKY family is conserved (Figure 2). Therefore, through multiple alignment analysis, it was found that the domain sequence of the WRKY family is also very conservative, mainly including the WRKYGQK signature sequence (Figure 2A) and β fold (Figure 2B). The WRKY domain has approximately 60 amino acids, with the sequence WRKYGQK13-14-CX4-5CX22-23HXH (C2H2) or WRKYGQKX13-14-CX7CX23HXH (C2H2), and WRKYGQK typically becomes a signature sequence (Figure 2A). Group I contains two WRKY domains, with the second WRKY domain containing a PR intron insertion fragment located between the C2 and H2 sequences. Group II contains a WRKY domain with two intron insertion fragments, namely the PR intron and VQR intron. Group III contains a WRKY domain, as well as the PR intron and VQR intron insertion fragments, but the domain C-terminus contains the -HXC sequence, which is different from the other two sets of sequences.

The 3D structural analysis of the WRKY family shows that all three subgroups contain four β folds and distribute evenly from the N-terminus to the C-terminus (Figure 2B). The sequence and spatial structure of the WRKY family indicate that they are highly conserved among 24 species of angiosperms, with plants from different evolutionary positions sharing the same WRKY domain and β folding; therefore, it is necessary to further explore the biological functions of the *WRKY* family.

### 2.3. Origin and Evolution of the WRKY Family

The *WRKY* family is one of the largest transcription factor families in plants. The first *WRKY* gene was cloned from sweet potato, and then it was continuously cloned from other species, such as wild oats (*Avena fatua* L.), parsley (*Petroselenium crispum* L.), and *Arabidopsis thaliana* [1,2,4]. As for the origin of the WRKY gene, the research results show that it originated from eukaryotic organisms approximately 1.5 to 2 billion years ago, but it has not appeared in fungi and animals so far. Therefore, it is speculated that the *WRKY* gene exists exclusively in plants [2,6,7]. Compared with non-flowering plants, the WRKY transcription factor plays a more important role in flowering plants. In particular, the third subfamily *WRKY* gene, which is the fastest evolving *WRKY* gene with the best adaptability, is mainly formed after the differentiation of monocots and dicots; the number of *WRKY* families also changes with the evolution of plants [4,7].

We selected 24 species of angiosperms, including basal angiosperms, monocotyledonous, and dicotyledonous plants, for species evolution analyses, and the results of reconstructing the phylogenetic tree indicate that the evolution of the *WRKY* family is consistent with the evolutionary status of the species genome. Among them, the basal angiosperms include *Amborella trichopoda* and *Nymphaea colorata*, the magnolias include *Cinnamomum kanehirae*, *Liriodendron chinense*, and *Liriodendron tulipifera*, as well as six monocotyledonous plants and thirteen dicotyledonous plants.

The number of gene families is also related to the evolutionary status of a species. Overall, the lower the evolutionary status, the smaller the number of family members in a species. If a whole genome replication event occurs in a species, the number of gene family members would also increase. From the perspective of the evolutionary status of the species, the whole genome duplication (WGD) event is also related to the evolutionary status [13]. The higher the plant, the more genome replication events may occur. In the long-term process of species evolution, plants continuously adapt to the environment and evolve many functions. Many genes lose their function or only retain some functions after the occurrence of whole genome doubling events, resulting in different numbers of genes retained in plants and ultimately forming different numbers of genes.

Among the 24 species of angiosperms, the total number of *WRKY* families in *Amborella trichopoda* is at least 34 (Table 2). Only four plants experienced two WGD events, *Cinnamomum kanehirae* from Magnoliaceae [14], *Carya illinoinensis*, *Sinapis alba*, and *Vitis vinifera* from Dicotyledonous, among others, all experienced a WGD event. The frequency of genome duplication in dicotyledonous plants is higher than that in magnolia, monocotyledonous, and basal angiosperms, and the total number of *WRKY* families in most dicotyledonous plants is also higher than in other angiosperms. For example, the most members of the dicotyledonous plant family are *Gossypium hirsutum* with 238 genes, the least is *Cirus trifoliata* with 55 genes, the most members of the monocotyledonous plant family is *Zea mays* with 137 genes, the least is *Zostela marina* with 46 genes, the most members of the magnolia family is *Cinnamomum kanehirae* with 73 genes, the least are *Lirioderon chinense* with 44 genes, and the most members of the basal angiosperm family are *Nymphaea corata* with 65 genes. The minimum number is *Amborella trichopoda,* with 34 genes.

The total number of *WRKY* families in the 24 species of angiosperms and the results of the WGD events indicate that the genetic functional diversity of dicotyledonous plants may be more abundant than monocotyledonous, magnolia, and basal angiosperms, and may also be more adaptable to environmental changes. Therefore, further exploration of the functional differentiation of basal angiosperms, magnolias, monocots, and dicots can help to better understand the functional differences of the *WRKY* family.

To explore the relationship with the *WRKY* family of different species during the evolutionary process, 24 species of angiosperms were analyzed using the “timetree” website (http://www.timetree.org/, accessed on 19 March 2024). The results showed that there are significant differences in the differentiation time of species with different evolutionary positions (Figure 3). The functional differentiation of gymnosperms and angiosperms began around 196 MYA, while basal angiosperms and magnolia plants underwent functional differentiation around 175 MYA. By 170 MYA, magnolia and monocotyledonous plants became more capable of differentiation, gradually forming independent evolutionary clades. After relatively brief evolution, monocots and dicots further underwent functional differentiation at 160 MYA, forming two independent branching species. Through continuous evolution, different species in each branch also underwent functional differentiation at different evolutionary time points, forming a variety of species today.

Through the analysis of the expansion and contraction of the *WRKY* family, it is concluded that the number of expansions and contractions of plants with different evolutionary statuses was different, and the number of plants with the same evolutionary status was also different. For example, the basal angiosperm plant *Amborella trichopoda* had more shrinkage gene numbers than the expansion ones, and conversely, *Nymphaea colorata* had much more expansion gene numbers than the contraction ones. *Liriodendron tulipifera* and *Cinnamomum kanehirae* had a greater number of expansions than the shrinkage number, and *Liriodendron chinense* had fewer expansion numbers than the shrinkage ones.

### 2.4. The Environmental Selection Pressure of the WRKY Family

By conducting an environmental selection pressure analysis on the *WRKY* family genes, the results showed that most of them undergo positive selection during environmental changes, and as long as a small number of genes undergo negative selection, they may gradually lose their function in subsequent evolution, which may also affect the number of members of the gene family [2,3,4,5]. By combining these results with the previous analysis, it can be concluded that angiosperms undergo more whole genome replication events, more genes undergo positive selection during environmental changes, and ultimately obtain more family members [4,5,6].

After analyzing the *WRKY* family Ka/Ks value of the 24 species of angiosperms, it was found that the plants with different evolutionary statuses suffer from different environmental selection pressures [5,7], and the ratio of positive and negative selection is also different. As shown in Table 3, three plants showed positive selection, the other fourteen plants showed negative selection, and seven plants did not undergo synonymous or nonsynonymous substitution.

For basic angiosperms, eighteen genes undergo replacement in water lilies, but under environmental selection conditions, only six genes are positively selected, indicating that the environment would eliminate unfavorable genes and retain favorable ones. No base substitution occurred in the genes of *Amborella trichopoda*, indicating that the *WRKY* gene sequence was stable and conserved during evolution, and no mutations occurred. The selection pressure of magnolia plants is different from that of basal angiosperms. The *WRKY* gene of the camphor tree did not undergo base substitution, indicating a relatively stable evolutionary process. *Liriodendron chinense* and *Liriodendron tulipifera* are different species of magnolia plants, but they exhibit completely different adaptations in environmental selection. Five genes of *Liriodendron chinense* undergo base substitution, while six genes of *Liriodendron tulipifera* undergo base substitution. However, *Liriodendron chinense* undergoes purification selection from the environment, while *Liriodendron tulipifera* undergoes positive selection from the environment. This indicates that during the evolutionary process, species of the same evolutionary status, due to differences in their living environment, experience differences in the evolution of the *WRKY* family, ultimately affecting the number of members of the *WRKY* family.

The environmental selection pressure of the *WRKY* gene varied more among different species of dicots, most of which were purified selection; only a few were positive selection, and some plants did not undergo base substitution. For example, no gene substitution occurred for *Corymbia citriodora*, *Carya illinoinensis*, *Coffea arabica*, and *Portulaca amilis*. *Arabidopsis thaliana*, *Cirus trifoliata*, and *Populus trichocarpa* all suffered positive selection, and the other plant *WRYK* genes all suffered from environmental purification selection. The *WRKY* gene of the monocots and dicots was subjected to a similar pattern of environmental selection pressure, and most plants were subjected to environmental purification selection; only a few of them had positive selection or no base substitution. For example, *Sorghum bicolor* suffered from positive selection, *Brachypodium distachyon* did not undergo base substitution, and plants such as *Oryza sativa* and *Zea mays* suffered from purification selection.

The above results showed that the base substitution of the *WRKY* gene in most plants was mainly subjected to the purification selection of the environment and the elimination of unfavorable genes, and only a few plants had a positive selection of the environment for base substitution. In some plants, the *WRKY* gene was relatively stable and did not undergo base substitution. Therefore, plants with different evolutionary positions also have genes that undergo base substitution in the process of adapting to the environment and eliminate unfavorable genes after environmental selection to achieve better evolution.

The number of single-copy genes varied among the species of different evolutionary statuses. The gene tree analysis of single-copy genes in the *WRKY* family showed that single-copy genes were also relatively conserved (Appendix A and Table 3). Overall, there were a total of 331 single-copy genes in the 24 species of angiosperms, with *Amborella trichopoda* having only six genes and *Sinapis alba* having thirty-three genes. Compared with the evolutionary tree grouping of the *WRKY* family genes, group I had the highest number of single-copy genes, while the other subgroups had little or almost no single-copy genes.

### 2.5. Development Function of WRKY Transcription Factor

The diversity of this gene family in plants is related to the life cycle of plants and their response to environmental stress. At present, the research on the *WRKY* gene has mainly focused on the growth and development of plants, and the response to resistance, such as seed germination, and the response to abiotic stresses, such as drought and low temperature. Studies have shown that the *WRKY* gene can regulate plant resistance through over-expression or gene knockout, which is also conducive to revealing the signal pathway mediated by the *WRKY* gene.

The WRKY transcription factor mainly regulates the expression of specific target genes and is subject to environmental factors and biological stress (such as bacteria, fungi, viruses, and other pathogens) or abiotic stress (such as exogenous hormones, high temperature, low temperature, high salt, mechanical injury, etc.), and the expression is specific in different tissues. The signal transduction pathways involved in WRKY regulation include plant hormone salicylic acid (SA), abscisic acid (ABA), jasmonic acid (JA), and the enzyme calmodulin (CaM).

The *WRKY* gene also regulates the growth, development, and reproductive senescence of plants, such as seed development, dormancy, germination, flowering, and the senescence of plants, and further regulates the growth, development, and metabolism of plants. In terms of biosynthesis, the 42 *GmWRKY* genes of *Gentiana macrophylla* can participate in secoiridoid biosynthesis, promoting the accumulation of secondary metabolites [15]. The *Lagerstroemia indica* L. contains 61 *LiWRKY* genes involved in regulating anthocyanin biosynthesis [16]. Among the 72 *JsWRKY* genes of *Jasminum sambac*, the overexpression of *JsWRKY51* can enhance the accumulation of β-ocimene, which regulates the synthesis of aromatic hydrocarbon components [8]. In tomatoes, *SlWRKY75* can maintain auxin homeostasis and promote plant resistance [17]. In rose petals, *RhWRKY30* can promote the expression of the *RhCAD* gene and enhance the biosynthesis content of lignin [18]. The transcriptome of *Melastoma dodecandrum* revealed that the *MedWRKY* gene is involved in growth and development and is highly expressed in roots and mature fruits [19].

The senescence process at the end of the plant life cycle is inevitable, and the *WRKY* gene plays a regulatory role in petal senescence. In terms of senescence, the transcription factor TgWRKY75 of the tulip activates the biosynthesis of ABA and SA, accelerating the petal senescence process of the tulip [20]. This indicates that the WRKY transcription factor can not only positively regulate the aging process, but also participate in this process as a negative regulatory factor. In general, it is particularly important to study the function of the *WRKY* gene in the whole process of plant growth, development, metabolism, and senescence.

### 2.6. Functions of the WRKY Family in Biotic Stress

Plants grow in the natural environment and adapt to the changing environment through their defense mechanisms. Systemic acquired resistance and induced systemic resistance are two important ways for plants to resist pathogens. When plants are invaded by pathogens, they will resist the invasion of pathogens through a series of physiological reactions, such as the waxy layer of their leaves and the secretion of their body surface. The plant resistance pathway is regulated by salicylic acid and jasmonic acid, which activate or inhibit the expression of disease resistance genes through signal transduction [11].

In tomatoes, the overexpression of *SlWRKY75* can reduce the content of IAA, reduce the expansion of auxin proteins, upregulate the expression of the PRs and NPR1 genes, and enhance potato resistance to the *Pseudomonas syringae* pv. *Tomato* (*Pst*) DC3000 pathogen [17]. The full-length transcriptome and RNA-seq analyses of sesame revealed that the *WRKY* gene is highly expressed in response to *Corynespora casicola* stress and is a hub gene in the coexpression regulatory network [21]. The domain of *Arabidopsis AtWRKY45* is susceptible to two pathogens, *Pseudomonas syringae* pv. *Pisi* and *Ralstonia pseudosolanacearum*, which specifically recognize anchoring, thereby inhibiting the pathogen immune response in Arabidopsis [22]. The *Cicer arietinum* L. *WRKY* gene responds to *Ascomycta rabiei* infection and exhibits differential expression trends with other transcription factors during the response process [23]. Recent research has proposed a new method to capture the binding transcription factors of the *SERRATE*(*SE)* genes through CRISPR-dCas9 (CASPA dCas9). The results showed that the transcription factors AtWRKY19 and AtPAR2 jointly promote the expression of the *SE* genes and enhance the pathogen resistance of *Arabidopsis* [24].

Insects or pathogens in the environment can cause damage to plant growth. Therefore, increasing the biological resistance of plants helps them better adapt to environmental growth. Whether it is crops, horticultural plants, or woody plants, these are all plants that are beneficial for human survival and development, and in-depth research on their molecular mechanisms is of great significance.

### 2.7. Functions of the WRKY Family in Abiotic Stress

Under abiotic stress, to resist the influence of an adverse environment through a series of physiological regulation and metabolic processes [25,26,27], plants can normally grow, forming a complex gene regulation network in which the WRKY transcription factor plays a very important role [28]. With the continuous development of biotechnology, the expression pattern of the *WRKY* family in response to abiotic stress was verified via RNA-seq, real-time quantitative PCR, and other technologies, and the regulation mechanism of the *WRKY* family was expounded [26].

The WRKY transcription factor regulates the response of plants to abiotic stress. For example, in wheat, TaWRKY transcription factors are involved in regulating the response to aluminum and manganese ion stress [29]. The transcriptome of low-phosphorus-stressed wheat revealed that *TaWRKY74s* is the major gene involved, which may regulate plant adaptation to low-phosphorus stress through ABA and auxin signaling [30]. In *Pyrus betulifolia*, the *PbWRKY* gene responds to both high temperature and drought stress and exhibits a high expression trend [31]. The transcriptome of *Gossypium anomalum* seedlings revealed that the *GaWRKY* gene is involved in salt stress response, and was validated via qRT-PCR to be consistent with RNA-seq data [32].

The 145 *TrWRKY* genes of the white clone respond to cold stress, and most genes are upregulated in the early stages of cold stress [33]. The 64 *DhWRKY* genes of *Dendrobium huoshanense* respond to hormone and low temperature stress. *DhWRKY42* significantly responds to jasmonic acid stress [34]. The *WRKY* family of *Liriodendron chinense* also participates in the abiotic stress response, and *LchiWRKY18* and *36* are involved in high temperature and drought stress responses, with high expression levels [35]. *StWRKY016*, *StWRKY045*, and *StWRKY055* in potatoes specifically respond to high-temperature stress in leaf tissues [7].

The *WRKY* gene family is involved in both biological and abiotic stress functions, regulating plant adaptation to different environments (Figure 4). Biological stress mainly includes insect stress, pathogen stress, and microbial stress. Abiotic stress mainly includes temperature, ion, and osmotic stress. In the long-term evolutionary process, the *WRKY* family exhibits diverse functions in adapting to environmental changes, and there are also differences in the biological functions of different subgroups. For example, most genes respond to abiotic stress, but have little response to the third group of *WRKY* members. In some species, the third group of *WRKY* members almost do not respond to abiotic stress but respond to biological stress. The *LchiWRKY18* and *36* genes of *Liriodendron chinense* are both members of the second group and participate in the response to high temperature and drought abiotic stress [35]. Combined with the promoters of downstream target genes *LchiHSPs* and *LchiMED26*, the expression of these target genes promotes plant response to abiotic stress environments.

Environmental stress includes many factors, including temperature, osmotic stress, and ion stress, which mainly affect plant growth [26,27]. Insect and pathogen stress is mainly biological stress that occurs during a certain stage of plant growth and changes in the external environment (Figure 4). In the living environment of angiosperms, stimulated by environmental factors, cells in the body begin to respond to these environmental stresses. The receptors on the cell membrane perceive environmental stimuli and transmit signals to various signaling pathways within the cell, such as ROS, hormones, ABA-dependent substances, and WRKY transcription factors. These substances then feed the signals back to various WRKY transcription factors in the nucleus for a response. These transcription factors further bind to the promoter sequences of downstream target genes, activating the expression of various downstream target genes. After the expression of these target genes, various response mechanisms are activated, such as stomatal closure, leaf shrinkage, the production of osmotic protective substances, and the activation of enzyme system responses, promoting plant adaptation to environmental changes.

## 3. Discussion

### 3.1. The Evolution of the WRKY Family in Higher Plants Is Relatively Conservative

Plants also undergo whole genome duplication events during their evolutionary process, and the number of occurrences varies among plants of different evolutionary statuses [28,36,37]. Overall, there are more plants with dicotyledonous leaves that experience WGD twice than other plants, while monocotyledonous plants and basal angiosperms rarely experience WGD events twice [36,37]. Magnolia plants only have camphor trees experiencing WGD events twice [14]. Most plants undergo a WGD event and retain the replicated genes to form the current total number of *WRKY* families.

*WRKY* also undergoes multiple functional differentiation during the purification process, with functional differentiation occurring in gymnosperms and angiosperms around 196 MYA. By 175 MYA, the basal angiosperms had undergone functional differentiation with magnolia and monocotyledonous plants, forming three branches of plant evolution. After a long period of evolution, around 160 MYA, monocotyledonous and dicotyledonous plants underwent functional differentiation, forming new plant branches.

Compared to the evolutionary statuses and functional differentiation times of plant genomes, the functional differentiation time of basal angiosperms and magnolias *WRKY* family is earlier and slower than that of the genome (Appendix A). The evolutionary status of the *WRKY* family in magnolias and basal angiosperms is the same, which may be due to differences in the evolutionary process between the genomes and individual families.

### 3.2. The Environment Selection Preserves the Existing WRKY Number of Family Members and Different Expansion–Contraction Ratios

Environmental selection eliminates unfavorable genes in plants and preserves favorable genes, so the analysis of environmental selection can better understand the replacement of gene bases in plants during the process of adapting to environmental changes, and thus understand the stability of plant genes [38,39]. Most genes in plants undergo environmental selection during their evolutionary process. The *WRKY* selection pressure analysis from basal angiosperms to monocotyledonous plants shows that most genes undergoing base substitution are subjected to environmental purification selection. Only some plant genes undergo positive selection, and dicotyledonous plants have more positive selection than monocotyledonous plants.

The expansion and contraction of gene families are also key factors affecting the number of families. Further analysis of the expansion and contraction of the *WRKY* family in angiosperms shows that most plants experience more expansion than contraction, while some plants experience less expansion than contraction. Many species of monocotyledonous plants may experience contraction, and only *Zea mays* has a greater expansion than contraction. Magnolia plants are relatively unique, with only the number of contracted of *Liriodendron chinense* greater than the number of expanded ones. Combined with the total number analysis of the *WRKY* family, the number of *Liriodendron chinense* is less than the other two species. These results indicate that the total number of *WRKY* family members is not only related to the number of WGD events but also the degree of environmental selection and expansion–contraction, ultimately determining the total number of family members.

### 3.3. The Conservatism of Domain and 3D Structure of the WRKY Family

The WRKY domain is mainly specifically bound to the cis-regulating element W-box (T) (T) TGAC (C/T) sequence of the target gene promoter to regulate the expression of the downstream target gene. The core conserved sequence of this cis-regulating element is TGAC, which is also the key sequence that the WRKY protein can specifically bind to [1,4,9]. Through the bioinformatics and functional analysis of the promoter, it is concluded that in the stress-related promoter sequence, the W-box, is generally combined into the promoter via clustering [40]. Among the genes related to disease resistance and aging, the W-box mainly starts the expression of these genes, and the WRKY transcription factor is mainly involved in the response of these processes in the response of plants to environmental stress [3,4,5]. In addition, some experiments have shown that the WRKY transcription factor also has a regulatory effect on plant growth and development. Some WRKY proteins may participate in seed development and ABA-mediated growth inhibition after germination, regulating the formation of leaf hair and the senescence process.

The three-dimensional structure of a protein determines its function. The WRKY family consists of four main components, which are β-fold proteins. Its main characteristic sequence is WRKYGQK, which also contains two types of intron insertion sequences, the PR intron and VQR intron. The former is mainly inserted after the WRKYGQK sequence, while the latter is mainly inserted after the second C base of C-C. According to the analysis of multiple sequences in the *WRKY* family of basal angiosperms, magnolias, and monocotyledonous plants, it can be concluded that *WRKY* is relatively conserved in family evolution.

The first subgroup of the WRKY family in THE 24 species of angiosperms contains two WRKY domains, and PR intron insertion mainly occurs in the second WRKY domain sequence. The second and third subgroups both contain a WRKY domain, and both intron insertion sequences, the PR intron and VQR intron, appear in these two subgroups. The three-dimensional structure of *WRKY* is also very conservative, with four north tower folds in the three-dimensional structures of different subgroups across the 24 species of angiosperms. The above results indicate that the *WRKY* family has a very stable grouping quantity and sequence structure during the evolutionary process, and plants of different evolutionary positions have the same function.

### 3.4. Functional Diversity of the WRKY Family in Angiosperms

The *WRKY* gene family is involved in many biological processes in plants, including biotic stress and abiotic stress [3,4,41]. Biological stress mainly refers to insect, pathogen, and microbial stress, while abiotic stress mainly includes temperature, osmotic, and ion stress [3,41,42]. When plants adapt to different environments, the receptors on the cell membrane first sense the external stimuli, stimulate changes in various physiological indicators of the cytoplasm, stimulate the expression of transcription factors in the nucleus, and then combine with downstream target gene promoters to induce the expression of target genes, promoting plant adaptation to various environments [4,12,41].

The overexpression of the splicing variant *AtrWRKY42-2* in *Amaranthus Paniculatus* enhances the expression of the *AtrCYP76AD1* gene and increases the biosynthesis of betaine [43]. In tulips, the overexpression of *TgWRKY75* enhances the expression of the *TgNCED3* gene, increases ABA and SA biosynthesis, and jointly promotes the leaf senescence process [20]. Under calcium ion stress, the *SlWRKY* gene in potatoes participates in both the positive and negative regulation of the stress response, promoting crop adaptation to heavy metal environments [44]. In wheat, silencing the *TaWRKY31* gene leads to poor plant growth status and poorer resistance to drought stress. The overexpression of the *TaWRKY31* gene in *Arabidopsis thaliana* can enhance its drought resistance, reduce water loss rate, and reduce stomatal opening [45]. The overexpression of the maize *ZmB12D* gene in *Arabidopsis thaliana* enhances its waterlogging tolerance. Enzymatic hybridization experiments have shown that *ZmB12D* interacts with *ZmWRKY70*, regulating maize waterlogging resistance [46]. The *PmWRKY70* gene was cloned from the *Prunus mume* cultivar “Guhong zhusha”, and the overexpression of *PmWRKY57* enhanced the cold stress resistance of *Arabidopsis thaliana* plants [47].

The *WRKY* family is involved in plant regulation with group specificity, with the majority of abiotic stresses being mainly mediated by the second group. The first and third groups are primarily involved in biological stress or other tissue growth and development processes. For example, in magnolia plants, the *WRKY* family of *Liriodendron chinense* is mainly the group II involved in high temperature and drought stress, and *LchiWRKY18* (II-e) and *LchiWRKY36* (II-d) reach their expression peaks at the 24-h and 72-h time points of stress [35]. The group IIc GhWRKY transcription factor in cotton enhances plant resistance to *Fusarium oxysporum* stress by mediating *GhMKK2* flavonoid synthesis [48].

In summary, the *WRKY* family is relatively conservative in terms of grouping and 3D structure in the evolution of angiosperms. However, in plants of different evolutionary positions, due to the different environmental adaptability of plants, different evolutionary processes, such as WGD events, environmental selection pressures, and family expansions and contractions ultimately result in differences in the number of *WRKY* family members among different plants, leading to various differential characteristics in adapting to the environment. This may be the underlying mechanism of plant diversity, and with the continuous development of science and technology and in-depth research, more functions of the *WRKY* family will continue to be explored.

## 4. Materials and Methods

Twenty-four WRKY family protein sequences from the plant genome website Phytozome(V13) (https://phytozome-next.jgi.doe.gov/, accessed on 19 March 2024) were used [49]. We found and downloaded the *WRKY* keyword in the search box, checked for no duplicate sequences, and then used OrthoFinder2 software to analyze the species evolution status of the *WRKY* family [50,51]. We organized the *WRKY* sequences of each species into a “Fasta” format file, and then placed the “Fasta” files of 24 species in the same folder. Then, we ran the command line “orthofinder2 -f folder” to obtain the results. We used the default parameter “Fasttree” in the program to construct the species tree. For all the other parameters, we used the default parameters in the software [50,51]. The time of species functional differentiation was analyzed on the “time tree” website (http://www.timetree.org/, accessed on 19 March 2024) and was displayed at the bottom of the evolutionary tree [52].

The construction of the phylogenetic tree was first carried out using ClustalX (v2.1) software to perform the multiple sequence alignments and save them into the “Fasta” format. Then, the evolutionary tree was constructed using BEAST (v2.6.6) software. The “Fasta” format files were converted into XML files using the BEAUTi (v2.6.6) program and the site model was set to Dayoff. Then, the BEAST (v2.6.6) program was imported for 10,000,000 MCMC sampling to construct a Bayesian evolutionary tree [53]. Finally, the parameter burning percentage was selected to 90 through the TreeAnnotator (v2.6.6) program and the posterior probability limit was set to 1 [53]. Then, we checked for low memory, annotated the evolutionary tree, and used Figtree (v1.4.3) to obtain the evolutionary tree and the posterior values of all the branches, which were displayed with two decimal places on the evolutionary tree.

The 3D structural analysis was conducted through the online website NCBI (https://www.ncbi.nlm.nih.gov/Structure/CN3D/cn3d.shtml, accessed on 19 March 2024), which imported the protein sequences from the *WRKY* family to obtain the PDB files. Then, a 3D browsing program Cn3D (v4.3.1) was downloaded from the website to visualize the spatial structure and obtain the 3D structural maps. The domain sequence conservatism of the *WRKY* family was obtained through multiple sequence alignment analysis using ClustalX (v2.1). The conservatism of the WRKY domain was then segmented and visualized by integrating the conservatism of different sequence fragments. The Ka and Ks values were calculated using KaKs_Calculator_3.0 (V3.0), and the select pressure was measured according to the Ka/ks value [54,55,56].

## 5. Conclusions

During the process of adapting to the environment, plants undergo changes in their genes, enzyme systems, and biological pathways to better adapt to the environment. WRKY transcription factors are involved in the various life activities of plants. Although the number varies among different angiosperms, they are generally divided into three major groups and seven subgroups. In terms of evolution, WRKY’s domain is relatively conservative and contains WRKYQGK signature sequences. The tertiary structure of the WRKY protein is highly conserved across different subgroups, each containing four β-folds. Due to the varying number of replication events throughout the entire genome and the different environmental choices experienced during evolution, different plants ultimately retain different numbers of *WRKY* family members. According to the evolutionary results of the *WRKY* family in basal angiosperms, magnolia plants, and monocotyledonous and dicotyledonous plants, the higher the evolutionary status, the relatively more members of the *WRKY* family, and the better their potential for adapting to environmental changes.

The human living environment is deteriorating, and global warming and rising temperatures are constantly breaking new records. Plants constantly adapt to new high temperature and drought environments, and the expression and response of WRKY transcription factors in the body are crucial for plants to survive in high temperature and drought environments. There are also various other environmental factors, such as phosphorus stress, low temperature stress, and salt stress, which can limit plant growth. Therefore, studying the *WRKY* gene family is crucial for revealing the biological mechanisms of plant adaptation to the environment.

With the rapid development of science and technology and the maturity of biotechnology, such as genetic manipulation, more and more research will continue to emerge. The molecular mechanisms of plant environmental stress will become more in-depth, and the functional diversity of the *WRKY* gene family will become clearer with the deepening of scientific research.

## Figures and Tables

**Figure 1 ijms-25-03551-f001:**
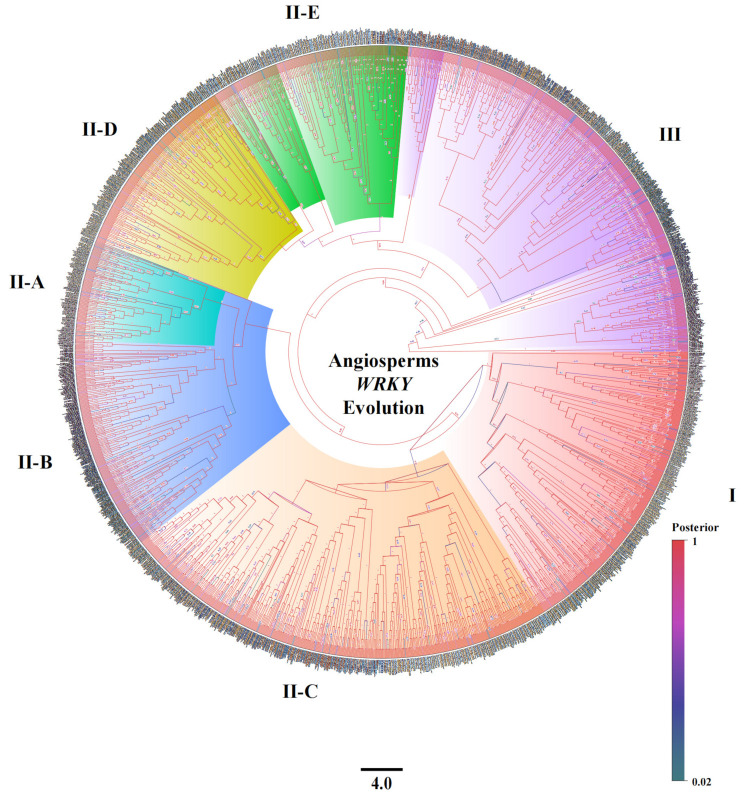
Twenty-four species of angiosperms from the WRKY evolutionary tree. A total of 2274 WRKY genes are divided into three major groups and seven subgroups, I, II-A II-E, and III, with the different colors representing different subgroup branches. The bottom 4.0 represents the branch length scale, and the posterior represents the Bayesian posterior value of the branch.

**Figure 2 ijms-25-03551-f002:**
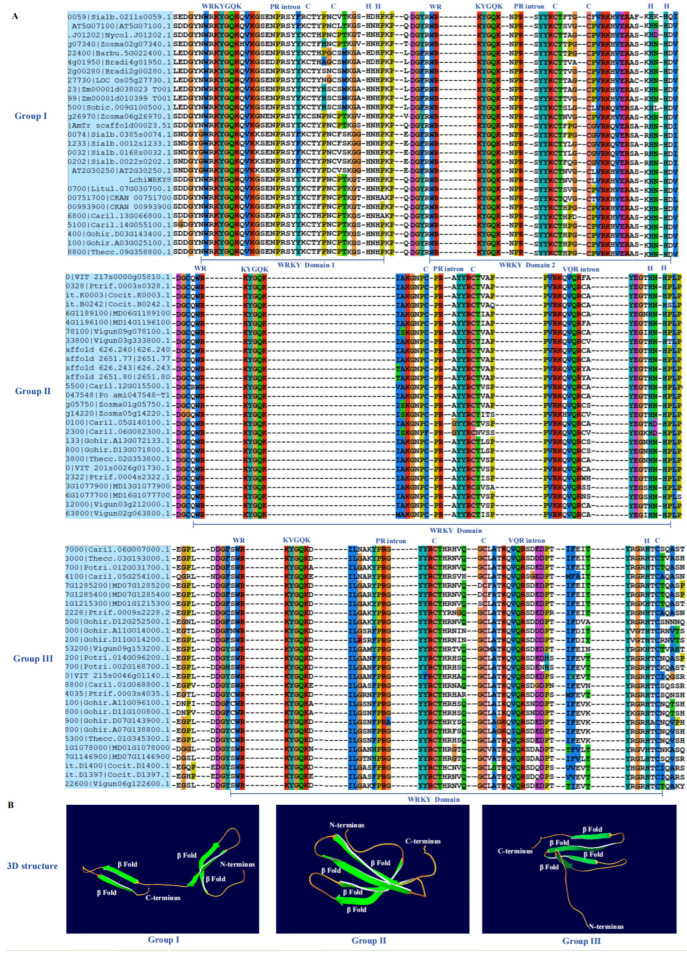
Domain sequence conservation and 3D structure of the WRKY family. (**A**) The sequences of the three subgroups of WRKY all contain WRKYGQK signature sequences, with two types of PR intron and VQR intron insertions. Group I contains two WRKY domains, while the other subgroups only contain one WRKY domain. Different colored backgrounds represent different base conservatism. (**B**) The three-dimensional spatial connectivity domains of the three subgroups all contain four β folds.

**Figure 3 ijms-25-03551-f003:**
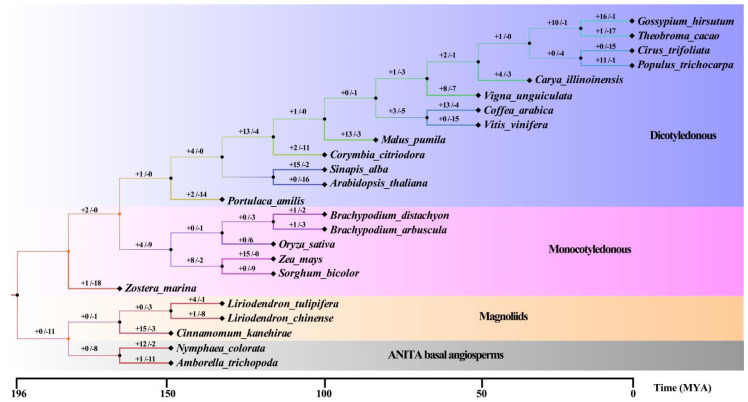
The evolutionary statuses and functional divergence times of 24 species of angiosperms. The numerical values on the branch represent the number of expansions and contractions, + represents expansion, - represents contraction, and the bottom coordinates represent the species divergence time, and different colored backgrounds represent different evolutionary branch plant categories.

**Figure 4 ijms-25-03551-f004:**
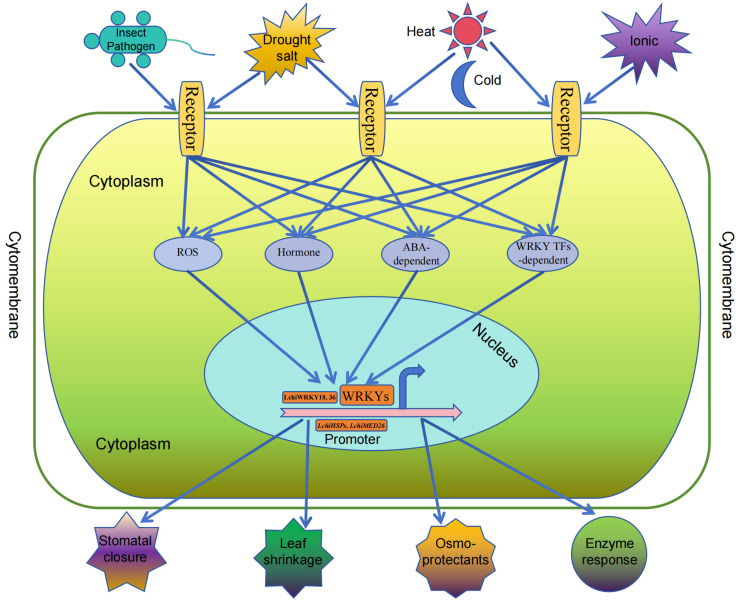
*WRKY* functional diversity summary chart. The different shapes in the upper part represent different external environmental stress conditions, the middle part represents the overall cell, and the bottom represents the different responses after stress.

**Table 1 ijms-25-03551-t001:** Number of members in different subgroups of 24 species of angiosperms.

Species	Group I	Group II-A	Group II-B	Group II-C	Group II-D	Group II-E	Group III
*Amborella trichopoda*	7	2	5	6	3	3	8
*Nymphaea colorata*	18	6	9	10	7	10	5
*Cinnamomum kanehirae*	15	6	8	12	9	12	11
*Liriodendron chinense*	7	3	6	8	4	7	9
*Liriodendron tulipifera*	9	6	7	13	6	7	10
*Brachypodium arbuscula*	13	4	8	23	7	10	26
*Brachypodium distachyon*	17	5	5	16	10	10	26
*Oryza sativa*	7	4	8	19	6	11	51
*Sorghum bicolor*	9	5	8	20	7	11	36
*Zea mays*	15	7	12	30	12	17	44
*Zostera marina*	7	2	8	10	6	6	7
*Arabidopsis thaliana*	12	3	8	18	7	7	28
*Carya illinoinensis*	17	6	13	24	9	10	12
*Cirus trifoliata*	10	3	8	13	5	7	9
*Coffea arabica*	23	5	15	28	13	11	32
*Corymbia citriodora*	16	6	11	18	5	8	15
*Gossypium hirsutum*	36	16	31	69	28	26	32
*Malus pumila*	23	6	15	25	14	13	28
*Populus trichocarpa*	23	5	9	24	13	12	14
*Portulaca amilis*	12	5	5	14	10	8	21
*Sinapis alba*	41	8	26	64	24	19	51
*Theobroma cacao*	10	3	8	14	6	6	18
*Vigna unguiculata*	15	6	15	22	7	11	22
*Vitis vinifera*	14	3	8	15	9	6	12

Note: The different background colors represent different classifications of plants; from top to bottom, they were basal angiosperms, Magnolia, Monocotyledonous, and Dicotyledonous. The table is arranged in order based on the first character of the plant Latin scientific names and different colored backgrounds represent different evolutionary branch plant categories.

**Table 2 ijms-25-03551-t002:** Total number of *WRKY* families and whole genome replication events in 24 species of angiosperms.

Species	Family	Taxonomy	WGD	Total Number
*Amborella trichopoda*	Amborellaceae	ANITA Basal angiosperms	1	34
*Nymphaea colorata*	Nymphaeaceae	ANITA Basal angiosperms	1	65
*Liriodendron chinense*	Magnoliaceae	Magnoliids	1	44
*Liriodendron tulipifera*	Magnoliaceae	Magnoliids	1	58
*Cinnamomum kanehirae*	Lauraceae	Magnoliids	2	73
*Zostera marina*	Zosteraceae	Monocotyledonous	1	46
*Brachypodium distachyon*	Poaceae	Monocotyledonous	1	89
*Brachypodium arbuscula*	Poaceae	Monocotyledonous	1	91
*Sorghum bicolor*	Poaceae	Monocotyledonous	1	96
*Oryza sativa*	Poaceae	Monocotyledonous	1	106
*Zea mays*	Poaceae	Monocotyledonous	1	137
*Cirus trifoliata*	Rutaceae	Dicotyledonous	1	55
*Theobroma cacao*	Malvaceae	Dicotyledonous	1	65
*Vitis vinifera*	Vitaceae	Dicotyledonous	2	67
*Portulaca amilis*	Portulacaceae	Dicotyledonous	1	75
*Corymbia citriodora*	Myrtaceae	Dicotyledonous	1	79
*Arabidopsis thaliana*	Brassicaceae	Dicotyledonous	1	83
*Carya illinoinensis*	Juglandaceae	Dicotyledonous	2	91
*Vigna unguiculata*	Fabaceae	Dicotyledonous	1	98
*Populus trichocarpa*	Salicaceae	Dicotyledonous	1	100
*Malus pumila*	Rosaceae	Dicotyledonous	1	124
*Coffea arabica*	Rubiaceae	Dicotyledonous	1	127
*Sinapis alba*	Brassicaceae	Dicotyledonous	2	233
*Gossypium hirsutum*	Malvaceae	Dicotyledonous	1	238

Note: WGD stands for whole genome duplication event, and the total number represents the total number of *WRKY* family members per species. The table is arranged in ascending order based on the total number of plants in different classifications and different colored backgrounds represent different evolutionary branch plant categories.

**Table 3 ijms-25-03551-t003:** Analysis of environmental selection pressures in the *WRKY* family.

Species	Taxonomy	Gene Pairs	Ka/Ks < 1	Ka/Ks > 1	Selection Pressure	Singe Copy
*Amborella trichopoda*	ANITA Basal angiosperms	0	0	0	\	6
*Nymphaea colorata*	ANITA Basal angiosperms	18	12	6	Negative	10
*Cinnamomum kanehirae*	Magnoliids	0	0	0	\	9
*Liriodendron chinense*	Magnoliids	5	3	2	Negative	7
*Liriodendron tulipifera*	Magnoliids	6	2	4	Positive	7
*Brachypodium arbuscula*	Monocotyledonous	105	57	48	Negative	14
*Brachypodium distachyon*	Monocotyledonous	0	0	0	\	15
*Oryza sativa*	Monocotyledonous	229	140	89	Negative	13
*Sorghum bicolor*	Monocotyledonous	157	67	90	Positive	11
*Zea mays*	Monocotyledonous	366	201	165	Negative	19
*Zostera marina*	Monocotyledonous	7	5	2	Negative	7
*Arabidopsis thaliana*	Dicotyledonous	18	5	13	Positive	11
*Carya illinoinensis*	Dicotyledonous	0	0	0	\	15
*Cirus trifoliata*	Dicotyledonous	7	3	4	Positive	8
*Coffea arabica*	Dicotyledonous	0	0	0	\	21
*Corymbia citriodora*	Dicotyledonous	0	0	0	\	12
*Gossypium hirsutum*	Dicotyledonous	115	71	44	Negative	32
*Malus pumila*	Dicotyledonous	27	21	6	Negative	17
*Populus trichocarpa*	Dicotyledonous	18	8	10	Positive	18
*Portulaca amilis*	Dicotyledonous	0	0	0	\	13
*Sinapis alba*	Dicotyledonous	55	30	25	Negative	33
*Theobroma cacao*	Dicotyledonous	7	4	3	Negative	9
*Vigna unguiculata*	Dicotyledonous	15	9	6	Negative	13
*Vitis vinifera*	Dicotyledonous	9	8	1	Negative	11

Note: Gene pairs represent the number of genes that underwent base substitutions, and the number 0 indicates that the sequences without genes in the species underwent substitutions. The table is arranged in order of the first character of the plant Latin scientific names and different colored backgrounds represent different evolutionary branch plant categories.

## Data Availability

The data in this article has been publicly published and can be downloaded from the website and the Appendix A provided in the article.

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
