# Peer review of "Evolution of the *WRKY* Family in Angiosperms and Functional Diversity under Environmental Stress"

_ijms, 2024, doi:10.3390/ijms25063551_

Round 1

Reviewer 1 Report

Comments and Suggestions for Authors

The manuscript “Evolution and Functional Diversification of the WRKY Family in Angiosperms” deals about the role of WRKY family in response to various external stimuli in Magnoliaceae plants.

I think that some revisions are necessary:

In the introduction it is necessary to add some sentences about the role of WRKY and what is the goal of the review.

In the Results section you should explain how you selected the species to construct the tree.

The sentence “This result….families” (lines 81-84) is obvious, explain better your hypothesis.

Figure 2 is unclear; the characters is too small. Replace “Domian” with “Domain”.

The sentences in the lines 212-219 should be supported by references.

You should rewrite the conclusions, with future perspectives.

Comments on the Quality of English Language

Minor revisions are required

Reviewer 2 Report

Comments and Suggestions for Authors

Dear Editor,

The submitted review article, coauthored by Mr. Wu and his colleagues, is dedicated to the evolution of the WRKY family in angiosperms and its functions in responding to various external environments. The topic of this review could be quite interesting to potential readers, but the presented manuscript may lack relevance due to the absence of relevant findings and information. It is really basic and doesn’t really seem to respond to any new research questions.

I suggest the authors improve the manuscript and separate completely the information about the functions of the WRKY family in biotic and abiotic stress. Also, to change the scientific names of the botanical families (Cruciferae, Brassicaceae, Gramineae, Poaceae, etc.), to italicize the plant scientific names, and to reformulate some phrases mentioned in the attached manuscript.

Furthermore, the manuscript contains several typographic and grammatical mistakes that should be double checked. Detailed suggestions are presented in the attached manuscript.

Comments on the Quality of English Language

Minor correction.

Round 2

Reviewer 2 Report

Comments and Suggestions for Authors

In its present form, overall the manuscript is good written and the ideas presented by the authors are easy to follow, and the new facts brought to the readers can be understandable

Author Response

Response: Thank you very much for your recognition and support. I will continue to review and improve the article so that online readers can read and discuss academic topics together.